# Development and Psychometric Evaluation of the End-of-Life Nursing Competency Scale for Clinical Nurses

**DOI:** 10.3390/healthcare12161580

**Published:** 2024-08-08

**Authors:** Ji-yeon Kim, Hyun-sun Kim, Mi-jung Kang, Hee-young Oh, Mi-rae Jo

**Affiliations:** 1College of Nursing, Eulji University, 712, Dongil-ro, Uijeongbu-si 11759, Gyeonggi-do, Republic of Korea; jykim421@eulji.ac.kr (J.-y.K.); hoh123@eulji.ac.kr (H.-y.O.); 2Department of Nursing, Jeonbuk Science College, Jeongeup-si 56204, Jeonbuk-do, Republic of Korea; jofuture@jbsc.ac.kr

**Keywords:** end-of-life, nursing competency, psychometrics, factor analysis, statistical methods

## Abstract

This study aimed to develop and establish psychometric properties of the End-of-Life Nursing Competency Scale for Clinical Nurses. The initial items were derived from an in-depth literature review and field interviews. The content validation of these items was assessed over three rounds by experts in end-of-life nursing care. The study included 437 clinical nurses from four hospitals in S, E, and D cities in South Korea. The final exploratory factor analysis resulted in a scale consisting of 21 items with the following five factors that explained 68.44% of the total variance: Physical care—imminent end-of-life, legal and administrative processes, psychological care—patient and family, psychological care—nurses’ self, and ethical nursing. The final model with these five subscales was validated through confirmatory factor analysis. Both item convergent-discriminant validity and known-group validity, which compared two groups based on clinical experience (*p* < 0.008) and working department (*p* < 0.008), were satisfactory. The internal consistency, as measured by Cronbach’s α, ranged from 0.62 to 0.89 for the subscales and was 0.91 for the total scale. This scale has been validated as a reliable and effective instrument for clinical nurses to self-assess their end-of-life nursing competencies in a clinical setting.

## 1. Introduction

Death is an inevitable event; however, no one can experience their death or predict what will happen afterward, which makes emotions related to death complex and challenging [1,2]. As life draws to a close, not only do family members experience profound shock and a sense of loss, but nurses who care for dying patients also face significant emotional turmoil. Nevertheless, since everyone desires to face their final moments without pain and with their inherent human dignity intact, healthcare professionals, particularly nurses, play an essential role in alleviating the suffering of dying patients and assisting them in achieving a dignified death [3,4].

In modern society, the prevalence of nuclear families has led to an increase in the number of people dying in medical facilities. Historically, due to a cultural emphasis on family values, Koreans preferred to pass away at home. However, there has been a steady rise in the number of deaths occurring in medical institutions, which now account for 75.4% of all deaths—more than four times the number occurring at home in Korea [5]. Specifically, end-of-life (EOL) nursing care for terminal and chronic patients is often administered in the general units or intensive care units of large hospitals, which is attributed to a shortage of hospice services [5,6].

EOL nursing care refers to comprehensive services that address the physical, psychological, and spiritual needs of patients and their families, enabling them to approach life’s final moments with peace, dignity, and quality of life [7]. Nurses, as primary healthcare providers, play a crucial role in helping patients and their families comfortably prepare for and accept the inevitability of death [8]. Unlike traditional nursing, which primarily focuses on disease and treatment [9], the key responsibilities of nurses in EOL nursing care involve managing symptoms and pain associated with the underlying disease, providing psychosocial and spiritual support, caring for dying patients and their families through effective communication, facilitating optimal decision making, and addressing the ethical dimensions of palliative care [10].

The provision of EOL nursing care encompasses a range of caregiving activities delivered during the final stages of a patient’s life. The nurse’s competence significantly influences the quality of EOL nursing care [11]. High-quality EOL nursing care can alleviate anxiety and depression in dying patients and their families and facilitate a more comfortable acceptance of death [12]. However, nurses often encounter challenges in providing EOL nursing care, which may stem from inadequate communication or dealing with sensitive family caregivers. Additionally, nurses who repeatedly experience the deaths of patients with whom they have developed a rapport may suffer from stress and depression [13,14,15].

EOL nursing competency is a multidimensional concept that encompasses the knowledge, attitudes, and behaviors necessary to perform EOL nursing care effectively [16]. This competency includes various elements such as the nursing process, the nurse’s personality, responsibility, interpersonal relationships, communication, roles as educators, and nursing ethics [17,18,19,20]. Higher levels of EOL nursing competency in nurses are associated with reduced stress during EOL care and an improvement in professional quality of life. Therefore, it is crucial to ensure that nurses who care for dying patients possess adequate EOL nursing competency [11,21].

To improve nurses’ EOL nursing competency, it is essential to first assess their competency using an accurate measurement tool, and then develop and implement programs aimed at improving this competency. The tools presented in the previous literature for measuring EOL care performance have several limitations. For instance, the tool developed by Park and Choi [22] focuses solely on patient-centered care and excludes family members from the care focus. Its spiritual care components are tailored to specific religious practices, which may not be applicable to nurses caring for patients without a religious background. Additionally, the tool by Montagnini, Smith, and Balistrieri [16] is specifically designed for nurses in intensive care units, rendering it unsuitable for nurses in general units. The EOL care questionnaire by Montagnini et al. [23], while designed for all healthcare workers, is not tailored specifically for nurses, thereby limiting its usefulness in assessing nurses’ EOL nursing competency. Moreover, existing tools fail to adequately assess crucial clinical competencies, such as determining the stage of EOL based on the patient’s rapidly changing condition, providing family-centered care that includes the patient in imminent EOL situations, and addressing the nurse’s ethics and post-mortem care. Specifically, post-mortem care encompasses both the technical skills required to handle the recently deceased body and the psychosocial aspects of supporting and communicating with the bereaved family, discussing EOL care outcomes, and assisting with funeral arrangements [4]. 

Therefore, this study aimed to identify the specific EOL care needs during the imminent stages, confirm the key elements of EOL care in clinical settings based on previous research, and develop a new measurement tool, the End-of-Life Nursing Competency Scale for Clinical Nurses (ELNCS-CN), which incorporates these factors. ELNCS-CN is designed for nurses caring for patients approaching the EOL, and to confirm its reliability and validity. 

## 2. Materials and Methods

### 2.1. Study Design

This study involved methodological research that focused on developing a measurement tool to assess EOL nursing competency among clinical nurses, as well as establishing its reliability and validity.

### 2.2. Tool Development Process

The development of the tool was carried out in two primary phases: the tool development stage and the tool validation stage, adhering to the guidelines set forth by DeVellis and Thorpe [24]. This process incorporated the recommended tool development procedures and included a qualitative evaluation, culminating in a three-step approach. The initial stage focused on deriving components through a hybrid model [25] to pinpoint elements essential for EOL nursing competency. The subsequent stage involved the preliminary development of the tool, starting with the creation of initial items. This was followed by three rounds of content validity verification and a preliminary survey to refine the tool. The final stage employed statistical methods to validate the tool and finalize the measurement instrument.

#### 2.2.1. Step 1: Deriving Attributes

To identify the components and detailed attributes of clinical nurses’ EOL nursing competency, previous studies were reviewed. The selection of databases followed the Core, Standard, Ideal (COSI) model from the systematic literature review manual of the National Evidence-based Healthcare Collaborating Agency (NECA). The databases used included PubMed, Cumulative Index of Nursing and Allied Health Literature (CINAHL), Embase, Research Information Sharing Service (RISS), Korean Studies Information Service System (KISS), and Data Base Periodical Information Academic (DBpia), with the languages restricted to Korean and English. Literature was selected from January 2003 to July 2023. The keywords for the domestic databases were “end-of-life nursing” AND (“competence” OR “ability” OR “capability assessment” OR “competency evaluation”) AND (“tool” OR “instrument” OR “scale” OR “assessment tool”), and for international databases, the search terms selected through Medical Subject Headings (MeSH) were (“end-of-life nursing” OR “end-of-life care nursing” OR “end-of-life nursing competency” OR “end-of-life care competency” OR “palliative care”) AND (“competence” OR “ability” OR “capability assessment” OR “competency evaluation”) AND (“tool” OR “instrument” OR “scale” OR “assessment tool”). A total of 381 articles were retrieved through domestic and international databases, and 39 duplicate articles were excluded. The titles and abstracts of 342 articles were reviewed, excluding 303 articles that did not address end-of-life care, were not related to nursing competency, or did not focus on nurses. Six articles without full text and two articles in languages other than Korean or English were also excluded. Finally, 31 articles were reviewed. From the literature review, 27 attributes regarding nursing competency in EOL care were derived. 

To expand and concretize the concepts identified in the literature review with practical insights, semi-structured interviews were carried out. The participants selected for field verification included one professor with over three years of experience in an intensive care unit and eight nurses currently employed in the intensive care unit at a university hospital, chosen through purposive sampling with no known biases or conflicts of interest, ensuring voluntary participation and adherence to ethical standards. After the study’s purpose was explained and written consent was obtained, the interviews were conducted either individually or in groups, each lasting between 30 and 40 min. The primary questions posed during the interviews included the following: “Please describe a memorable experience you’ve had while providing nursing care for a patient in the terminal stage”; “What is the essential nursing care that should be provided to the patient or their family as death nears?”; “What is crucial knowledge for nurses caring for patients in the imminent EOL stage?”; “What is the most important attitude for a nurse to maintain when caring for a terminal patient?”; and “Can you recall any significant incidents resulting from a lack of nursing competency during the imminent EOL stage?” Interviews took place in a quiet and calm environment. They were recorded and subsequently transcribed for analysis. The number of field interviews was determined by data (attributes) saturation, referring to when collecting more data no longer yields any new data. A total of 28 attributes were derived by adding 1 attribute identified in the field interviews to the 27 attributes identified through the literature review. Supporting References for the final End-of-Life Nursing Competency Items are listed in Appendix A.

#### 2.2.2. Step 2: Preliminary Tool Development

(1)Construction of Preliminary Items and Determination of the Tool’s Scale

Based on the 28 attributes of essential EOL care services identified in the literature review and field verification stages, along with the conceptual attributes of “competency”—namely, knowledge, attitude, and behavior [26]—50 preliminary items were constructed. The tool utilized a 5-point Likert scale format (1 = “strongly disagree”, 2 = “disagree”, 3 = “neutral”, 4 = “agree”, and 5 = “strongly agree”), which included a neutral option to accommodate respondents who may not have a definitive attitude.

(2)Expert Content Validity Verification

Content validity was assessed using both the content validity index (CVI) and subjective qualitative evaluation. Initially, three nursing professors conducted a qualitative evaluation of the preliminary items for the first content validity assessment. For the second round of content validity verification, seven experts were consulted, including two nursing professors with ICU experience, four geriatric nurse specialists, and one nursing professor with an ICU nursing certification, all possessing over three years of clinical experience. The item CVI (I-CVI) and the scale CVI (S-CVI) for the 50 items were calculated and verified according to Lynn’s criteria [27]. Each item was rated on a 4-point Likert scale for validity (1 = “not relevant”, 2 = “somewhat relevant”, 3 = “quite relevant”, and 4 = “very relevant”), and subjective feedback was gathered. The I-CVI for the second content validity test ranged from 0.43 to 1.00. Items with an I-CVI below 0.80 [28] and those considered similar in content were eliminated, resulting in the removal of 17 items and leaving a total of 33 items. For the third content validity verification, to incorporate the perspectives of the EOL care field, seven nurses with over 3 years of clinical experience in ICUs or general units at university or general hospitals were consulted. This verification found no items with an I-CVI below 0.80, indicating that no further revisions were necessary.

(3)Preliminary Survey and Finalization of the Tool

A survey was conducted to assess the face validity of the preliminary 33-item tool. The survey sample included 25 nurses who met the same criteria as the study participants and were selected through convenience sampling. These nurses were briefed on the study’s purpose, content, and procedures, and they provided written consent. The survey required participants to evaluate their understanding, the clarity and appropriateness of the terms used, and the time needed to complete the tool, using a 4-point scale. The findings indicated that the sentences were neither difficult nor ambiguous, and the completion time was under 15 min. The finalized version of the tool comprised 33 items, organized into sub-factors with the following item counts: physical care—imminent EOL (7 items), psychological care (17 items), legal and administrative processes (6 items), and ethical nursing (3 items).

#### 2.2.3. Step 3: Tool Validation and Finalization

(1)Study Participants

To validate the preliminary tool, we recruited a convenience sample of 437 participants. These participants were nurses employed at general or university hospitals in S, E, and D cities in South Korea. All participants had at least one month of clinical work experience and had provided written consent to participate in the study. 

Out of the 437 questionnaires received, 30 were excluded due to incomplete responses, leaving 407 for the final analysis. The determination of the sample size was guided by recommendations that 150–200 participants are suitable for exploratory factor analysis (EFA) [29], with a minimum of 150 participants required for confirmatory factor analysis (CFA) [30]. Furthermore, to ensure the consistency of the factor structure validated through EFA in subsequent verifications, it is advised to use different participants [29]. Consequently, using Excel for random sampling, 200 participants were designated for EFA and 207 for CFA.

(2)Research Tool

##### Self-Efficacy Related to Palliative Care

Convergent validity was assessed using a tool developed by Pfister et al. [31] to measure specific self-efficacy in palliative care. This instrument was designed for nurses in nursing homes who provide palliative and EOL care [32]. The tool was translated into Korean by Lee [32]. It includes 10 items that evaluate various aspects: symptom management and nursing care, social and psychological issues in nursing, counseling and support for the patient’s family, basic nursing care, and education about medications. Each item is scored on a Likert scale ranging from “strongly agree” (1 point) to “strongly disagree” (4 points), with total scores varying from 10 to 40. Higher scores reflect greater self-efficacy in palliative care. The tool’s reliability was confirmed with a Cronbach’s α of 0.86 during its initial development and 0.82 in Lee’s study [32], while it reached 0.88 in our study.

(3)Data Collection Method

This study received approval from the Institutional Review Board of E University (EU23-28) before commencement. Data collection occurred between 2 February 2024 and 9 April 2024. The nursing departments of one general hospital and three university hospitals in S, E, and D cities were contacted to discuss the study’s objectives and methods to secure approval for data collection. Recruitment postings were shared by the department managers in each hospital’s nursing department through mobile group chat rooms. Interested participants clicked on a URL that directed them to the questionnaire. There, they read an explanation of the study, completed the consent form, and filled out the questionnaire.

(4)Data Analysis

SPSS 23.0 and AMOS 23.0 (IBM Corp., New York, NY, USA) were used to assess the tool’s validity and reliability. The general characteristics of the participants were analyzed using descriptive statistics, including mean, standard deviation, frequency, and percentage. The item analysis was conducted by confirming normality through examining item means, standard deviations, skewness, and kurtosis. The contribution of each item to the total score was evaluated through item-total correlations. Additionally, skewed response distributions and ceiling and floor effects were checked, ensuring that the frequency of the lowest or highest scores was less than 30% [33].

Construct validity was analyzed using EFA and CFA. Validity was further verified through assessments of item convergent-discriminant validity, known-group validity, and discriminant validity. The EFA began with the evaluation of data suitability, employing the Kaiser–Meyer–Olkin (KMO) measure and Bartlett’s test of sphericity. The principal axis factor method was used to extract factors, followed by the application of the Promax rotation method [34]. Factors were selected based on eigenvalues exceeding 1.0 and a cumulative variance explanation of over 60%. Items were retained if they had factor loadings greater than 0.40 and cross-factor loading differences greater than 0.20 [35].

CFA was conducted to assess the fit of the model proposed by EFA. The evaluation of model fit was based on several criteria: a chi-square minimum/degree of freedom (CMIN/DF) ratio less than 3, a standardized root-mean-square residual (SRMR) and a root mean square error of approximation (RMSEA) less than 0.05 indicating good fit and 0.05 to 0.08 indicating acceptable fit, respectively; comparative fit index (CFI) and Tucker–Lewis index (TLI) values above 0.90; and lower Akaike information criterion (AIC) values indicating a better fit [34,36]. Additionally, standardized factor loadings greater than 0.50, average variance extracted (AVE) above 0.50, and construct reliability (CR) above 0.70 were examined to ensure convergence. Discriminant validity was assessed by confirming that the confidence intervals [Φ ± 2 × SE] of the correlation coefficients (Φ) did not include 1.0 [34]. Additionally, assessed by confirming that the Heterotrait–Monotrait Ratio of Correlations (HTMT) value should not exceed 0.85 [37]. Convergent validity was analyzed by examining the correlation between the developed tool and the specific self-efficacy related to palliative care instrument [31] using Pearson’s correlation coefficient. Known-group validity was assessed by analyzing differences in EOL nursing competency scores based on clinical experience or department using *t*-tests. To verify the reliability of the tool, internal consistency reliability was checked via Cronbach’s alpha > 0.70 [38] and McDonald’s omega > 0.70 [39]. All statistical significance probabilities were set at *p* < 0.05.

## 3. Results

### 3.1. General Characteristics of Participants 

For the participants included in the explanatory factor analysis (N = 200), the most common age group was 30–39 years, comprising 43.0% of the sample, and 94.0% of the participants were female. A majority, 76.5%, had earned a bachelor’s degree, and 54.5% reported no religious affiliation. In terms of clinical experience, the largest group (22.5%) had between 5 and 10 years of experience. Additionally, 73.0% of the participants were employed in university hospitals. The general units were the most frequent department of employment, representing 48.5% of the sample, and 81.0% of the participants held the position of staff nurse.

For the participants included in the CFA (N = 207), the age group of 20–29 years was the most represented, comprising 43.0% of the sample, and 88.9% of the participants were female. A majority, 79.2%, had earned a bachelor’s degree, and 57.0% reported no religious affiliation. In terms of clinical experience, the largest group (27.1%) had between 5 and 10 years of experience. Additionally, 62.8% of the participants were employed in university hospitals. The general units were the most common workplace, involving 50.7% of the participants, and 82.6% held the position of staff nurse (Table 1).

### 3.2. Item Analysis 

The item analysis was conducted on a total of 33 items. The mean scores for these items varied from 3.14 to 4.29. Skewness values ranged from −0.62 to 0.03, and kurtosis values from −0.54 to 1.29, both of which met the criteria for normality with skewness within 2 and kurtosis within 7 [40]. The floor effect for each item was between 0.0% and 4.4%, which satisfied the criterion of being less than 30% for all items [33]. The ceiling effect ranged from 4.9% to 40.3%, with two items (item 28 and item 29) exceeding 30% and were therefore reviewed and subsequently deleted. The correlation coefficients between the items and the total score ranged from 0.39 to 0.76, with all coefficients exceeding 0.30, confirming the suitability of the items [35]. Following the item analysis, 31 items were retained.

### 3.3. Validity Analysis

#### Construct Validity

(1)Exploratory Factor Analysis

After excluding 2 items based on the results of the item analysis, an initial EFA was conducted on a total of 31 items. This analysis yielded a Kaiser–Meyer–Olkin (KMO) value of 0.92 and Bartlett’s test of sphericity value of χ2 = 3951.53 (*p* < 0.001), confirming the appropriateness of the data for factor analysis. The analysis identified six factors with eigenvalues exceeding 1.0, which together accounted for 66.8% of the total variance. Following this, nine items were removed: one item (item 13) due to a factor loading below 0.40; seven items (items 5, 20, 22, 14, 21, 31, 25) because of cross-loadings with a difference in factor loadings of less than 0.20; and one item (item 32) that did not load onto any factor. Then, a second EFA was performed on the remaining 22 items.

The second EFA yielded a KMO value of 0.90 for 22 items and Bartlett’s sphericity test value of χ2 = 2553 (*p* < 0.001), identifying five factors that explained a cumulative variance of 68.5%. After removing one item (item 8) due to cross-factor loading, a third EFA was conducted on the remaining 21 items.

The third EFA produced a KMO value of 0.89 for 21 items and Bartlett’s sphericity test value of χ2 = 2327.47 (*p* < 0.001), identifying five factors that explained a cumulative variance of 68.4%. All factor loadings exceeded 0.40, and no items exhibited cross-loadings. As a result, five factors comprising 21 items were established (Table 2).

The derived factors were named to accurately reflect their respective themes. Factor 1 was labeled “physical nursing—imminent EOL”, factor 2 as “legal & administrative processes”, factor 3 as “psychological nursing—patient & family”, factor 4 as “psychological nursing—nurses’ self”, and factor 5 as “ethical nursing”.

(2)Confirmatory Factor Analysis

CFA was conducted to confirm the validity of the model structure, which included the five factors and 21 items derived from EFA (Figure 1).

In the first CFA, the initial model fit indices of the 21-item model were as follows: CMIN/df = 2.57, SRMR = 0.07, RMSEA = 0.09, CFI = 0.86, TLI = 0.86, and AIC = 564.28. The CFI and TLI values did not meet the criterion of being greater than 0.90. To improve the model fit, 3 items with factor loadings below 0.50 were removed [41] (items 7, 23, 30), leaving 18 items remaining. In the second CFA with the 18 items, the model fit indices were as follows: CMIN/df = 2.46, SRMR = 0.06, RMSEA = 0.08, CFI = 0.90, TLI = 0.90. Additionally, the AIC value decreased to 398.83 from the initial 564.28, indicating an improved model (Table 3).

(3)Convergent and Discriminant Validity of Items

For the final 18 items, the standardized factor loading values ranged from 0.54 to 0.91, all exceeding the criterion of 0.50, and the critical ratio (C.R.) values were above 1.96 for all items [34]. The AVE values ranged from 0.54 to 0.68, with values above the criterion of 0.50 for all factors, and the CR ranged from 0.70 to 0.91, meeting the criterion of 0.70 or above [34] (Table 4).

The analysis of item discriminant validity revealed that the values, calculated as ±2 times the standard error (SE) from the correlation coefficients (Φ), ranged from 0.33 to 0.87. This range satisfies the criterion that excludes the absolute value of 1 [34]. HTMT values ranged from 0.44 to 0.78, satisfying the criterion of being less than 0.85 [37] (Table 5).

(4)Known-Group Validity

Known-group validity was assessed by examining differences in palliative care competencies based on the participants’ clinical experience and their department of work. The mean scores for palliative care competencies were 61.61 (±9.32) for those with less than three years of experience and 64.49 (±10.29) for those with three years or more, demonstrating a significant difference (*t* = −2.67, *p* = 0.008). Similarly, the mean scores varied by department, with ICU & ER staff scoring 65.54 (±8.73) and staff from other departments scoring 62.82 (±10.50), also showing a significant difference (*t* = 2.68, *p* = 0.008) (Table 6).

(5)Convergent Validity

To validate convergent validity, we examined the correlation between the scores from the newly developed ELNCS-CN and the self-efficacy scores related to palliative care. The analysis revealed a correlation coefficient of −0.56 (*p* < 0.001), indicating a moderate correlation and confirming convergent validity.

### 3.4. Reliability

To evaluate the internal consistency reliability of the nurse compensation measure-ment tool developed in this study, we calculated Cronbach’s α coefficient and McDonald’s omega coefficient. The tool’s overall Cronbach’s α was 0.91. The Cronbach’s α for the individual factors were as follows: factor 1 (physical nursing—imminent EOL) scored 0.89, factor 2 (legal & administrative processes) scored 0.83, factor 3 (psychological nursing—patient & family) scored 0.81, factor 4 (psychological nursing—nurses’ self) scored 0.73, and factor 5 (ethical Nursing) scored 0.62. The tool’s over-all McDonald’s omega was 0.92.

### 3.5. Tool Optimization and Finalization 

The tool optimization stage, the final step in the 8-step tool development process described by DeVellis and Thorpe [24], involved a qualitative evaluation that focused on the comprehension and accuracy of each item. This evaluation was conducted in collaboration with a nursing professor experienced in tool development. The items that were validated for validity and reliability were not changed. Ultimately, the ELNCS-CN was finalized with five factors and 18 items: physical care—imminent EOL (Managing physical symptoms, end-of-life stages, post-mortem care, and providing essential care: 5 items), legal & administrative processes (Executing legal responsibilities, using end-of-life services, and following protocols: 4 items), psychological nursing—patient & family (Providing emotional and spiritual support, counseling, and empathizing with patients and families: 4 items), psychological nursing—nurses’ self (Managing personal emotions and stress, handling death calmly, and caring for end-of-life patients: 3 items), and ethical nursing (Ensuring ethical care, reporting violations, and maintaining patient dignity and privacy: 2 items). Following the optimization stage, the complete tool is presented in Appendix A. 

## 4. Discussion

EOL nursing care is an essential task that every nurse should be equipped to perform. Given the increasing number of older adults and the rising prevalence of complex chronic diseases, more patients are now facing EOL situations in hospitals rather than at home. Consequently, it is anticipated that clinical nurses will increasingly encounter EOL scenarios [42]. Therefore, clinical nurses must possess competence in EOL care, assess their own skills, and address any deficiencies. To this point, an important resource is an EOL nursing competency assessment tool. Unlike tools designed for long-term settings such as hospice or palliative care, there is a need for assessment tools tailored for acute care settings, where nurses, patients, and families confront the immediacy of impending death. In this context, we discuss the key issues identified during the development and validation of the ELNCS-CN.

The EOL nursing competency tool developed in this study comprises five sub-factors and 18 items. Given its manageable number of items, it is anticipated to be readily utilized as a self-assessment tool by clinical nurses who are often burdened with heavy workloads. All items in the final tool satisfy the criteria for both ceiling and floor effects, demonstrating that there is no extreme bias in the responses. This feature enhances the tool’s effectiveness in measuring the intended concept.

Using the 21 items, the EFA revealed a total explanatory power of 68.44%, surpassing the 60% criterion suggested by Hamed [43]. In the field of social sciences, cumulative variance ratios between 50 and 60% are regarded as indicative of good explanatory power [44]. All factor loadings exceeded 0.40, demonstrating a satisfactory correlation between each item and its respective factor. The results of the CFA satisfied all established criteria, thereby validating each factor’s effectiveness in measuring EOL nursing competency.

Factor 1, “physical care—imminent EOL”, encompasses five items. It involves evaluating the patient’s rapidly changing conditions and identifying EOL stages to provide appropriate nursing care at each phase [45]. Essential tasks within this factor include providing nursing care tailored to the patient’s needs during the imminent EOL stage and managing and guiding post-mortem care [46]. This factor demonstrated the greatest explanatory power among the five factors, accounting for 39.96%.

Factor 2, accounting for 9.37% of the explanatory power, relates to the legal and administrative processes involved in EOL nursing care. This factor evaluates the awareness of institutional protocols for EOL care, understanding of legal and administrative processes, and the ability to perform nursing duties following advance directives [47].

Factor 3 is termed “psychological care—patient & family”. This factor encompasses four items that assess the psychological support provided to patients and their families, a crucial aspect as the EOL stage approaches. The period of imminent death underscores the importance of psychological care more than at any other time [47]. During these moments, the emphasis on spiritual care also becomes pronounced. Nurses are required to acknowledge and support the heightened emotions of grief, fear, and despair experienced by both patients and their families as death nears. Furthermore, a nurse’s facial expressions, tone of voice, and manner of speaking, which convey empathy towards the patient or family, are paramount [48]. The explanatory power of the third factor was 7.53%.

The fourth factor is “psychological care—nurses’ self”, with an explanatory power of 6.51%. When caring for patients who are facing imminent death, nurses must manage their fears, feelings of rejection, and tendencies to avoid the situation [49]. Additionally, the ability to control emotional agitation following the sudden, unexpected death of a patient is important in EOL nursing care [50].

The fifth factor is “ethical nursing”, which includes two items. In EOL care, adherence to the principles of biomedical ethics is crucial [51]. Izumi et al. highlighted the importance of maintaining patient privacy, promoting beneficence, preventing harm, and reporting any bioethical violations to an ethics committee in this context. The explanatory power of this factor was 5.08%.

The convergent validity of the items was confirmed, as the C.R. value exceeded 1.96 and the AVE value met the established criteria, demonstrating satisfactory correlations among the items that constitute the factors of non-compliance with self-care. The CR value surpassed the threshold of 0.50, ensuring construct validity. Discriminant validity was established for all factors, as the values obtained by adding and subtracting twice the SE from the correlation coefficients (Φ) did not include the absolute value of 1, indicating that the five sub-factors independently measured the concepts. Known-group validity was assessed with the hypothesis that nurses with longer clinical experience and those working in critical care units or emergency rooms, as opposed to general units, would exhibit higher EOL nursing competency. The analysis revealed statistically significant differences between the groups, thereby confirming known-group validity. This suggests that the tool can effectively measure differences between groups based on empirical phenomena grounded in the hypothesis. The verification of convergent validity demonstrated a statistically significant correlation between EOL nursing competency and self-efficacy in EOL nursing care, confirming that the tool is capable of measuring the correlation between these two convergent concepts.

Internal consistency reliability assesses how consistently all items of a tool measure a construct. This was confirmed with a total internal consistency reliability of Cronbach’s α = 0.91 and McDonald’s omega = 0.92, which meet the established criteria and ensure reliability. The internal consistency reliability of the subscales was satisfactory, with Cronbach’s α values of 0.70 or higher for factors 1 to 4. However, the reliability value for factor 5 was 0.62, failing to meet the criteria. The α value is influenced by the number of items, and a smaller number of items can lead to lower internal consistency reliability [52]. The lower internal consistency reliability of factor 5, which is below 0.70, is attributed to it consisting of only two items. In the exploratory factor analysis (EFA), many items showed high cross-loadings on multiple constructs. This is often due to the high interrelatedness of the items, as various aspects of EOL nursing competency overlap. The multifaceted nature of the competencies likely led to some items being relevant across multiple constructs. Additionally, certain items were excluded due to not meeting statistical parameters. Despite their relevance, they showed reduced importance and inconsistent responses in imminent death situations, impacting reliability and validity.

This tool stands out from existing instruments by specifically measuring the competencies required in imminent EOL nursing care situations. It features 18 items, which makes it concise and straightforward, with a completion time of under 15 min, thus ensuring it is convenient to administer. It avoids using terms or sentences that assess abstract concepts and focuses exclusively on evaluating nursing competencies relevant to imminent EOL nursing. Another benefit of this tool is that it assesses knowledge, attitude, and behavior evenly, reflecting the conceptual attributes of competence.

The EOL nursing competency measurement tool developed in this study has been demonstrated to be both valid and reliable, making it appropriate for assessing the competency of clinical nurses in Korea in providing EOL nursing care. Through self-assessment, individual nurses can identify their competency levels, which aids in delivering high-quality nursing care. Self-assessment is also believed to contribute to the development and application of educational programs aimed at improving deficiencies in EOL care competencies. When a nurse identifies areas of incompetence, immediate steps such as notifying supervisors, targeted training, and mentorship can be taken. Long-term measures include continuous professional development, regular assessments, and support programs, which can help address and improve these competency gaps. At the institutional level, efforts should be made to regularly implement educational programs to enhance nurses’ EOL care competencies and to incorporate these efforts into policy. Additionally, self-assessment can help determine the level of EOL nursing competency among nurses, assisting in the creation and execution of educational programs designed to improve their skills in this area. This study not only advances the theoretical understanding of EOL nursing competencies but also provides a practical framework for assessment and improvement in nursing education and practice. This study defines key competencies, provides a reliable assessment tool, and supports educational frameworks. It offers a comprehensive model for EOL nursing, enhancing competency evaluation and informing policy and best practices in nursing education and care.

The limitations of this tool are as follows. First, it is widely accepted that at least three items are necessary to obtain stable results from a single factor [44]. However, in this study, the factor concerning ethical nursing includes only two items, potentially compromising the stability of the results and resulting in lower internal consistency reliability. Second, the study’s scope was restricted to a limited number of research sites and utilized convenience samples, which hampers the generalizability of the findings and restricts their applicability to other populations. Additionally, the literature review was restricted to sources in Korean and English, which may have resulted in the omission of relevant studies published in other languages. Despite these limitations, the study has notable strengths. A thorough literature review identified essential attributes for EOL nursing competency. The sample size ensured reliable findings, and the confirmatory factor analysis (CFA) showed excellent fit with all factor loadings exceeding the acceptable threshold. 

Future research should aim to confirm the validity and reliability of the tool by involving nurses from a variety of workplaces, including hospitals, nursing homes, and other care facilities. Furthermore, using advanced statistical methods can provide deeper insights into the tool’s structural relationships and predictive validity. Additionally, evaluating item quality with techniques like item response theory (IRT) is recommended to enhance the tool’s precision and validity.

## 5. Conclusions

This study developed and validated a measurement tool designed for clinical nurses to self-assess their EOL nursing competency. This tool reflects the specific characteristics of the clinical environment in Korea and the essential components required for caring for patients nearing the end of life. The final version of the tool includes 18 items, organized on a 5-point Likert scale, and encompasses five factors: “physical care—imminent end of life”, “legal & administrative processes”, “psychological care—patient & family”, “psychological care—nurses’ self”, and “ethical nursing” (Appendix A). The tool’s construct validity was established through both exploratory and confirmatory factor analyses. Additionally, its convergent validity, known-group validity, and reliability were thoroughly assessed, confirming its effectiveness for measuring the EOL nursing competency of clinical nurses. This tool can be employed in future research focusing on nurses’ competence in EOL care. In a practical nursing context, it also allows nurses to self-assess their competency before engaging in the care of patients approaching the end of life.

## Figures and Tables

**Figure 1 healthcare-12-01580-f001:**
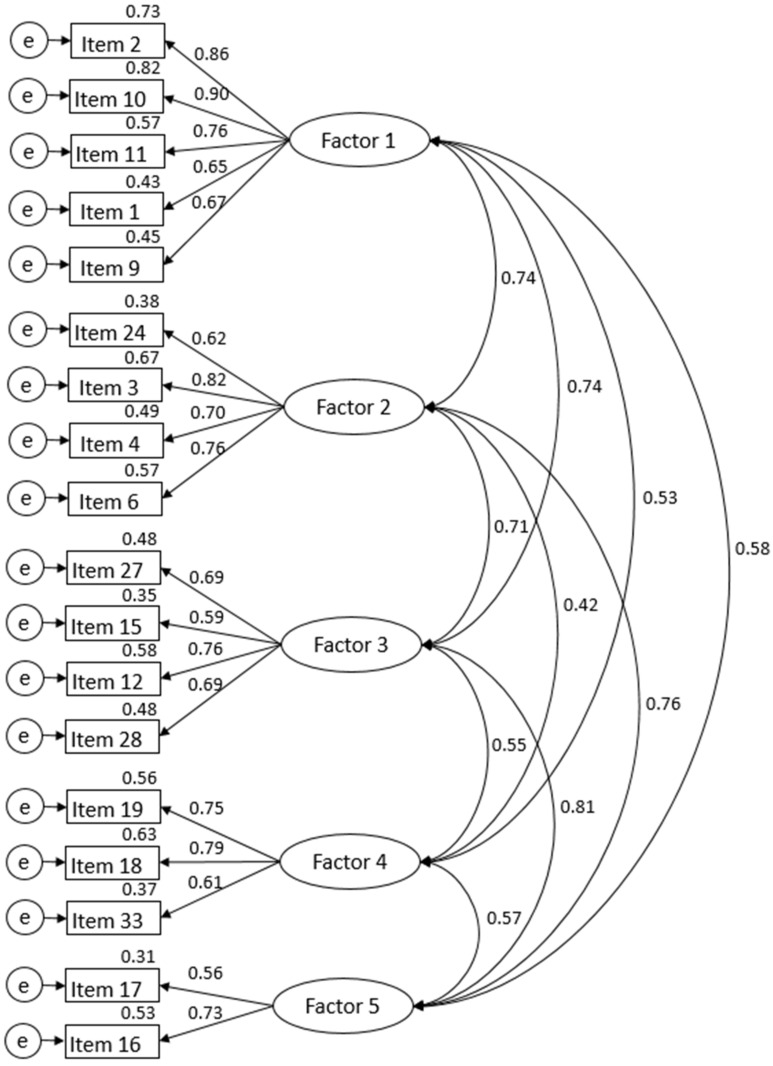
Findings of confirmatory factor analysis for the ELNCS-CN.

**Table 1 healthcare-12-01580-t001:** General characteristics of participants (N = 407).

Characteristics	Categories	Total (N = 407)	EFA Set (N = 200)	CFA Set (N = 207)
		N (%) or M ± SD	N (%) or M ± SD	N (%) or M ± SD
Age (year)	20~29	172 (42.3)	83 (41.5)	89 (43.0)
	30~39	167 (41.0)	86 (43.0)	81 (39.1)
	40~49	60 (14.7)	27 (13.5)	33 (15.9)
	≥50	8 (2.0)	4 (2.0)	4 (1.9)
		31.62 ± 7.06	31.68 ± 7.01	31.56 ± 7.18
Gender	Female	372 (91.4)	188 (94.0)	184 (88.9)
	Male	35 (8.6)	12 (6.0)	23 (11.1)
Level of education	Associate	10 (2.4)	7 (3.5)	3 (1.5)
	Bachelor’s	317 (77.9)	153 (76.5)	164 (79.2)
	Master’s	80 (19.7)	40 (20.0)	40 (19.3)
Religion	Yes	175 (43.0)	91 (45.5)	84 (40.6)
	No	232 (57.0)	109 (54.5)	123 (59.4)
Clinical experience (years)	<1	53 (13.0)	28 (14.0)	25 (12.1)
	1–<3	71 (17.4)	31 (15.5)	40 (19.3)
	3–<5	48 (11.9)	29 (14.5)	19 (9.2)
	5–<10	101 (24.8)	45 (22.5)	56 (27.1)
	10–<15	60 (14.7)	35 (17.5)	25 (12.1)
	≥15	74 (18.2)	32 (16.0)	42 (20.3)
Type of hospital	General	131 (32.2)	54 (27.0)	77 (37.2)
	University	276 (67.8)	146 (73.0)	130 (62.8)
Department	General unit	202 (49.6)	97 (48.5)	105 (50.7)
	ICU	106 (26.0)	57 (28.5)	49 (23.7)
	ER	13 (3.2)	6 (3.0)	7 (3.4)
	OPD	20 (5.0)	10 (5.0)	10 (4.8)
	Others	66 (16.2)	30 (15.0)	36 (17.4)
Position	Floor nurse	333 (81.8)	162 (81.0)	171 (82.6)
	Charge nurse	44 (10.8)	24 (12.0)	20 (9.7)
	Head nurse	30 (7.4)	14 (7.0)	16 (7.7)

CFA = Confirmatory factor analysis; ER = Emergency room; EFA = Exploratory factor analysis; ICU = Intensive care unit; M = Mean; OPD = Outpatient department; SD = Standard deviation.

**Table 2 healthcare-12-01580-t002:** Final exploratory factor analysis results (N = 200).

Factor	Item Number	Structure Matrix					Pattern Matrix					Communality
		1	2	3	4	5	1	2	3	4	5	
Physical Care—Imminent End-of-life	2	0.89	0.43	0.40	0.31	0.41	0.95	−0.10	0.08	0.06	−0.11	0.81
	10	0.87	0.40	0.31	0.35	0.48	0.91	−0.13	−0.05	0.10	0.03	0.77
	11	0.85	0.53	0.31	0.24	0.40	0.84	0.11	−0.03	−0.02	−0.07	0.73
	1	0.83	0.48	0.51	0.28	0.57	0.78	0.01	0.11	0.01	−0.07	0.64
	9	0.80	0.45	0.40	0.26	0.39	0.69	−0.01	0.19	−0.05	0.15	0.74
Legal & Administrative Processes	24	0.45	0.85	0.28	0.26	0.39	−0.08	0.86	0.01	0.03	0.07	0.74
	23	0.68	0.79	0.25	0.22	0.38	−0.30	0.83	0.36	0.02	−0.21	0.64
	3	0.61	0.76	0.31	0.23	0.41	0.41	0.62	−0.08	−0.04	−0.04	0.73
	4	0.62	0.73	0.10	0.22	0.53	0.25	0.61	0.01	−0.02	0.03	0.63
	6	0.20	0.69	0.42	0.16	0.12	0.28	0.55	−0.29	−0.04	0.29	0.69
Psychological Care—Patient & Family	27	0.39	0.29	0.84	0.26	0.38	0.05	−0.01	0.81	0.03	0.02	0.72
	15	0.61	0.42	0.77	0.20	0.48	−0.15	−0.02	0.65	0.09	0.31	0.71
	12	0.54	0.54	0.77	0.26	0.37	0.18	0.28	0.64	0.00	−0.10	0.62
	26	0.29	0.25	0.73	0.33	0.52	0.34	0.03	0.62	−0.11	0.08	0.71
Psychological Care—Nurses’ Self	19	0.24	0.25	0.05	0.84	0.23	0.04	0.10	−0.20	0.89	−0.09	0.75
	18	0.27	0.36	0.32	0.76	0.37	−0.12	0.20	0.10	0.70	0.04	0.62
	33	0.29	0.15	0.24	0.72	0.36	0.09	−0.13	0.03	0.68	0.09	0.53
	7	0.30	0.13	0.37	0.69	0.30	0.12	−0.16	0.22	0.66	−0.03	0.54
Ethical Nursing	30	0.35	0.22	0.41	0.22	0.84	−0.10	−0.11	0.14	−0.11	0.92	0.75
	17	0.53	0.45	0.46	0.33	0.80	−0.03	0.06	−0.05	0.21	0.70	0.68
	16	0.42	0.37	0.29	0.47	0.77	0.06	0.10	0.13	−0.02	0.68	0.63
Eigenvalue		8.39	1.97	1.58	1.37	1.07						
Explained variance (%)		39.96	9.37	7.53	6.51	5.08						
Cumulative variance (%)		39.96	49.33	56.85	63.36	68.44						

Kaiser–Meyer–Olkin = 0.89, Bartlett’s test of sphericity = 2327.47, *p* < 0.001.

**Table 3 healthcare-12-01580-t003:** Model fit of confirmatory factor analysis (N = 207).

Model	CMIN/df	SRMR	RMSEA	CFI	IFI	AIC
Model 1 (21 items)	2.57	0.07	0.09	0.86	0.86	564.28
Model 2 (18 items)	2.46	0.06	0.08	0.90	0.90	398.83

AIC = Akaike information criterion; CFI = Comparative fit index; CMIN = χ2 test; df = Degree of freedom; IFI = Incremental fit index; RMSEA = Root mean square error of approximation; SRMR = Standardized root mean residual.

**Table 4 healthcare-12-01580-t004:** Convergent validity of the scale (N = 207).

Factors	Items	Standardized Estimate (λ)	S.E.	C.R.	AVE	CR
Physical Care—Imminent End-of-life	2	0.67	0.06	5.30	0.68	0.91
	10	0.65	0.09	6.01		
	11	0.76	0.04	5.27		
	1	0.90	0.08	4.32		
	9	0.86	0.05	4.82		
Legal & Administrative Processes	24	0.76	0.04	9.41	0.56	0.83
	3	0.70	0.03	9.49		
	4	0.82	0.05	8.94		
	6	0.62	0.02	6.05		
Psychological Care—Patient & Family	27	0.69	0.03	7.59	0.60	0.85
	15	0.76	0.05	8.02		
	12	0.59	0.05	8.65		
	26	0.70	0.05	6.85		
Psychological Care—Nurses’ Self	19	0.61	0.06	9.17	0.57	0.79
	18	0.79	0.03	8.46		
	33	0.75	0.05	7.53		
Ethical Nursing	17	0.73	0.04	9.17	0.54	0.70
	16	0.59	0.04	8.42		

AVE = Average variance extracted; CR = Construct reliability; C.R. = Critical ratio; S.E. = Standard error.

**Table 5 healthcare-12-01580-t005:** Discriminant validity of the scale (N = 207).

Factors	HTMT					φ ± 2 × SE ≠ 1				
	F1	F2	F3	F4	F5		Estimate	SE	−2 × SE	+2 × SE
F1						F1↔F2 F1↔F3 F1↔F4	0.74	0.05	0.64	0.83
F2	0.76					F1↔F3 F1↔F3 F1↔F4	0.74	0.03	0.67	0.80
F3	0.69	0.62				F1↔F4 F1↔F3 F1↔F4	0.53	0.04	0.46	0.61
F4	0.50	0.44	0.49			F1↔F5 F1↔F3 F1↔F4	0.58	0.03	0.52	0.64
F5	0.65	0.72	0.78	0.60		F2↔F3 F1↔F3 F1↔F4	0.71	0.04	0.63	0.79
						F2↔F4 F1↔F3 F1↔F4	0.42	0.04	0.33	0.50
						F2↔F5 F1↔F3 F1↔F4	0.77	0.05	0.68	0.86
						F3↔F4 F1↔F3 F1↔F4	0.55	0.03	0.48	0.61
						F3↔F5 F1↔F3 F1↔F4	0.81	0.03	0.74	0.87
						F4↔F5 F1↔F3 F1↔F4	0.57	0.04	0.50	0.65

F1 = Physical Care—Imminent End-of-life; F2 = Legal & Administrative Processes; F3 = Psychological Care—Patient & Family; F4 = Psychological Care—Nurses’ Self; F5 = Ethical Nursing; SE = Standard error.

**Table 6 healthcare-12-01580-t006:** Known-group validity of the scale (N = 407).

Variables	Range of Score	n (%)	M ± SD	*t* (*p*)
Clinical Experience (years)	<3	124 (30.5)	61.61 ± 9.32	−2.67 (0.008)
	≥3	283(69.5)	64.49 ± 10.29	
Department	ICU & ER	119 (29.2)	65.54 ± 8.73	2.68 (0.008)
	Others	288 (70.8)	62.82 ± 10.50	

ER = Emergency room; ICU = Intensive care unit; M = Mean; SD = Standard deviation.

## Data Availability

The data used to support the findings of this study are available from the corresponding author upon request.

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
