# Peer review of "Development and Psychometric Evaluation of the End-of-Life Nursing Competency Scale for Clinical Nurses"

_healthcare, 2024, doi:10.3390/healthcare12161580_

Round 1
Reviewer 1 Report
Comments and Suggestions for Authors
The article is relevant and makes use of correctly applied methods. I leave some suggestions to improve the article:
1- The authors should present, through a table, the items that make up the Instrument and which references support its creation.
2- Before proceeding with the Exploratory Factor Analysis, it would be interesting to check the reliability of the data using McDonald's Omega;
3- The authors could explain why some items were excluded. The literature has indicated that the content of the items is relevant. So, why did the items not reach the statistical parameters to be maintained in the Instrument?
4- Is it also necessary to explain why so many items showed high cross-loadings on several constructs? Do you know if the content of the items was poorly formulated?
5- It is necessary to explain the content of the items that form each of the five constructs. How do the contents relate to generating these five constructs?
6- It was observed that the RMSEA value was 0.08. Authors such as Brown (2015) suggest a suitable model adjustment value occurs for RMSEA values ​​< 0.05. It would be necessary for authors to present more robust references for the RMSEA criterion equal to 0.08 to be considered valid;
References:
- Brown, T. A. (2015). Confirmatory factor analysis for applied research. Guilford publications.
7- After defining the final items, the authors could use McDonald's Omega and Cronbach's Alpha to reinforce their validation of the Instrument's final items.
8- The discriminant validity method needs to be stronger. I recommend the Fronell-Larcker criterion (Fornell; Larcker, 1981) or the Heterotrait-Monotrait Ratio of Correlations (HTMT) (Henseler et al., 2015).
References:
- Fornell, C., & Larcker, D. F. (1981). Structural equation models with unobservable variables and measurement error: Algebra and statistics.
- Henseler, J., Ringle, C. M., & Sarstedt, M. (2015). A new criterion for assessing discriminant validity in variance-based structural equation modeling. Journal of the academy of marketing science, 43, 115-135.
9- The authors could clarify this study's theoretical and practical implications.
10- The authors must leave suggestions for future work, including possibly analyzing the Instrument using PLS-SEM (Bianco et al., 2023). It would also be relevant to evaluate the quality of the items via Item Response Theory, suggesting analysis of the discrimination parameters of the final items and difficulty for the Item response alternatives (Barbosa et al., 2024).
References:
- Bianco, D., Bueno, A., Godinho Filho, M., Latan, H., Ganga, G. M. D., Frank, A. G., & Jabbour, C. J. C. (2023). The role of Industry 4.0 in developing resilience for manufacturing companies during COVID-19. International Journal of Production Economics, 256, 108728.
- Barbosa, A. S., Crispim, M. C., Da Silva, L. B., Da Silva, J. M. N., Barbosa, A. M., & Morioka, S. N. (2024). How can organizations measure the integration of environmental, social, and governance (ESG) criteria? Validation of an instrument using item response theory to capture workers' perception. Business Strategy and the Environment.
Author Response
Attached, please find our response file.

Reviewer 2 Report
Comments and Suggestions for Authors
Please see the attachment

Author Response

(The authors gave the same response as above.)

Reviewer 3 Report
Comments and Suggestions for Authors
Thank you for the opportunity to read your paper. I am attaching my suggestions in a document.

Author Response

(The authors gave the same response as above.)

Reviewer 4 Report
Comments and Suggestions for Authors
Development and Psychometric Evaluation of the End-of-Life Nursing Competency Scale for Clinical Nurses
Thank you for this opportunity to review this manuscript. The study is of very high quality in terms of its content and presentation. The researchers have developed a tool for assessing nurses’ end-of-life care competencies. The study was carried out methodically and logically. The items and factors in the tool appear to be sound and well-tailored for the nursing populations it has been made to assess. I commend the authors on their work and hope that the tool is translated into other languages in the future for use in other contexts. Below, I offer a few comments, all of them minor, for improvement of the manuscript.
I think it’s better to use “EOL” as an abbreviation for “end-of-life” following its first mention in the text.
Lines 87-93: Here you have two statements of purpose. “this study aimed to…” and “The purpose of this study was to…” I think it would be better to combine them.
Line 101: You need to add the names of those who created the guidelines. Change to: adhering to the guidelines set forth by DeVellis and Thorpe [24].
Line 190, 192. Please use the word “participants” rather than “subjects”
Line 205: Please remove “â‘ ” as it is not used in English.
Lines 208-209: There is only a partial sentence here: “translated the tool into Korean.”
Lines 219-221 and 227-234: Please change to the passive voice to be consistent with the rest of the methods section. E.g., “SPSS 23.0 and AMOS 23.0 (IBM Corp.) were used to assess the tool’s validity and reliability.”
Lines 315-316: Change to: “CFA was conducted to confirm the validity of the model structure, which included the five factors and 21 items derived from EFA (Figure 1).”
Line 384: I think this is overstated: “At this point, the critical resource…” “Change to “To this point, an important resource…”
Line 389: Here you mention for the first time that the ELNCS-CN is a self-assessment tool. Up to this point it has not been described in these terms. I would suggest, at line 389, to not refer to it as a self-assessment tool but rather a tool for assessing the end-of-life competencies of nurses. Then keep the explanation is lines 391-392, about how the tool is appropriate for use in self-assessment.
Line 423: Please put “psychological care – nurses’ self” in quotation marks.
Line 470. Change the colon after “follows:” to a period.
Line 490: Change to “…context, it also allows nurses…”
Line 491: Change: “end-of-life patients” to “patients approaching the end of life.”
Line 496: Remove “Please add:”
The references should follow the journal’s style.
Comments on the Quality of English Language
Development and Psychometric Evaluation of the End-of-Life Nursing Competency Scale for Clinical Nurses
Thank you for this opportunity to review this manuscript. The study is of very high quality in terms of its content and presentation. The researchers have developed a tool for assessing nurses’ end-of-life care competencies. The study was carried out methodically and logically. The items and factors in the tool appear to be sound and well-tailored for the nursing populations it has been made to assess. I commend the authors on their work and hope that the tool is translated into other languages in the future for use in other contexts. Below, I offer a few comments, all of them minor, for improvement of the manuscript.
I think it’s better to use “EOL” as an abbreviation for “end-of-life” following its first mention in the text.
Lines 87-93: Here you have two statements of purpose. “this study aimed to…” and “The purpose of this study was to…” I think it would be better to combine them.
Line 101: You need to add the names of those who created the guidelines. Change to: adhering to the guidelines set forth by DeVellis and Thorpe [24].
Line 190, 192. Please use the word “participants” rather than “subjects”
Line 205: Please remove “â‘ ” as it is not used in English.
Lines 208-209: There is only a partial sentence here: “translated the tool into Korean.”
Lines 219-221 and 227-234: Please change to the passive voice to be consistent with the rest of the methods section. E.g., “SPSS 23.0 and AMOS 23.0 (IBM Corp.) were used to assess the tool’s validity and reliability.”
Lines 315-316: Change to: “CFA was conducted to confirm the validity of the model structure, which included the five factors and 21 items derived from EFA (Figure 1).”
Line 384: I think this is overstated: “At this point, the critical resource…” “Change to “To this point, an important resource…”
Line 389: Here you mention for the first time that the ELNCS-CN is a self-assessment tool. Up to this point it has not been described in these terms. I would suggest, at line 389, to not refer to it as a self-assessment tool but rather a tool for assessing the end-of-life competencies of nurses. Then keep the explanation is lines 391-392, about how the tool is appropriate for use in self-assessment.
Line 423: Please put “psychological care – nurses’ self” in quotation marks.
Line 470. Change the colon after “follows:” to a period.
Line 490: Change to “…context, it also allows nurses…”
Line 491: Change: “end-of-life patients” to “patients approaching the end of life.”
Line 496: Remove “Please add:”
The references should follow the journal’s style.
Author Response

(The authors gave the same response as above.)
